



# CFD based design of diffuser augmented wind turbines

Ertem Vehid[1]

[1]MSc. Wind Energy,Statkraft

**Abstract.** Diffuser Augmented Wind Turbines (DAWTs) are promising regarding energy costs by increasing the energy output without changing the rotor size. The project focuses on both numeric investigations of the Generalized Actuator Disc (GAD)theory. This study aims to develop a new diffuser augmented wind turbine by following existing diffuser design concepts which are diffuser with single-element and diffuser with multi-elements. It consists of two main parts: the design of diffuser and the design of rotor. The numeric investigations for diffuser design were carried out on the Star-CCM+. Rotor geometries, which were designed with an edited version of Glauert Optimization Method, concluded the evaluation of the optimized diffuser designs by comparing the performance increase. A steady-BEM code, which was adopted to model diffuser augmented turbines, was used in the process. It is found that all the design options result in significant performance increases in comparison to the bare rotor. However, the single-element diffuser is the optimum design, among others, due to its high-performance increase with less design complexity.

## 1 Introduction

Lowering the cost of energy produced by wind turbines has been the most critical point for the development of technology and future designs. So far, the industry chose a path aiming more powerful wind turbines by improving the blade and tower design. The improvement processes mostly consist of increasing blade lengths and parallel to this, increasing tower heights. As a result of bigger rotor designs with higher towers, the environmental and social impacts of wind turbines have become an issue, Cîrstea (2015), despite the lower levelized cost of energy.

As an alternative way to increase power production with the same rotor size and tower height, the shrouded rotor concept is a promising method since it can even exceed the Betz-Jawceskesy limit of a bare rotor,Tavares Dias Do Rio Vaz et al. (2014).

The concept was discussed and investigated primarily based on power increase in respect of economics in the past by Oman and K. M. Foreman,Oman and Foreman (1973), and by Igra, who did on of the first experimental investigations of DAWT design, Igra (1981). Even though K. Forman argued that one of the main reasons to use DAWTs is to decrease the pay-back time of the investment, Foreman (1981), the ground-breaking success has not come for the concept yet.

In the past years, the academic interest for this concept seemed to increase again with the help of CFD tools running on more powerful computers. One of the main design challenges is the separation of flow, which lowers the efficiency of diffuser. The study aims to address this problem as maximizing the designed diffuser power output. Several design concepts for diffusers were examined initially, and three design concepts were determined, which are single-element diffuser, multi-element diffuser, and multi-element diffuser with a gurney flap.



For evaluation of three optimized diffuser designs, rotor geometries are optimized, And the evaluation is carried out with a steady BEM code edited for shrouded rotors.

## 2 Airfoil Selection

To be used in the diffuser design , 5 airfoil types were determined :

- S1223, Aranake et al. (2015)

- NACA4412,Igra (1981)

- Eppler E423, Aranake et al. (2015)

- GOE417A

- Eppler E63

GOE417A was included since it is a thin airfoil and has a more basic design than the other airfoils whereas Eppler E63 has a better aerodynamic characteristics in terms of maximum $\frac{C_l}{C_d}$ than other design options have.

### 2.1 Diffuser Coefficient

For the evaluation of airfoil profiles, diffuser coefficient($C_s$) which was derived from a mathematical model based on 1-D Momentum theory,Hjort and Larsen (2014). The model uses an unload case to calculate the diffuser coefficient which is why it was found to be more time-efficient than evaluating the diffuser characteristics based on power coefficient($C_p$) with different loading cases.

The control volume of the model and the equation of the diffuser coefficient are shown below on Fig.1 and Eq.1 respectively.

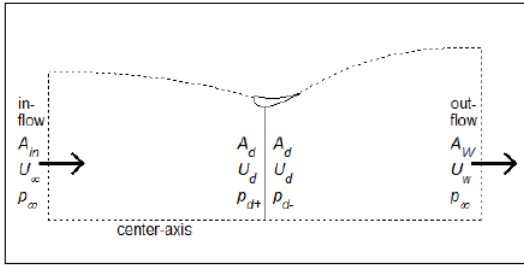

**Figure 1.** Control Volume of $C_s$ model

Hjort and Larsen (2014)

$$C_s = \frac{\sum_{i=1}^{n} \frac{A_d(i)}{A_d} U_d(i)}{U_\infty} - 1 \tag{1}$$



## 2.2 Evaluation

All simulations had the same mesh and domain size, and run with steady-state conditions. The diffuser angle and Re number were set as 0 and $1E+06$ respectively. The initial simulations did not indicate a clear difference between S1223 and Eppler E423 as it is shown at Tab. 1.

| Airfoil | $C_s$ | $\frac{C_L}{C_d}$ |
|---------|-------|-------------------|
| S1223 | 0.414 | 121.4 |
| Eppler E63 | 0.2142 | 235.2 |
| GOE417a | 0.1926 | 127.2 |
| Eppler E423 | 0.4251 | 156.5 |
| NACA4412 | 0.2441 | 129.4 |

**Table 1.** $C_s$ values with $\frac{C_L}{C_d}$ taken from Airfoiltools

In order to finalize the airfoil selection, a study on the diffuser angles (angle of attacks on an airfoil profile) was carried out for S1223 and Eppler E423 by comparing their diffuser coefficient values as well as checking the convergence of the simulations as it indicates the separation of flow along the diffuser surface.

### 2.2.1 Further Studies on Diffuser Angles

Diffuser angles of S1223 and Eppler E423 were increased with a step of 2-degrees starting from 0 degree. Fig.2 and Fig3 show
that the separation of flow along the trailing edge of the airfoil Eppler E423 resulted in a decrease on the diffuser coefficient. S1223, on the other hand reached a $C_s$ of 0.5 within the same angle range. Therefore, S1223 was determined as the base airfoil for diffuser designs.

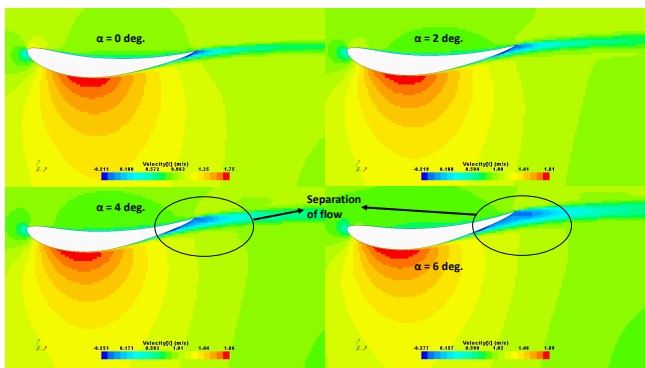

**Figure 2.** Separation of flow with different diffuser angles, Eppler E423



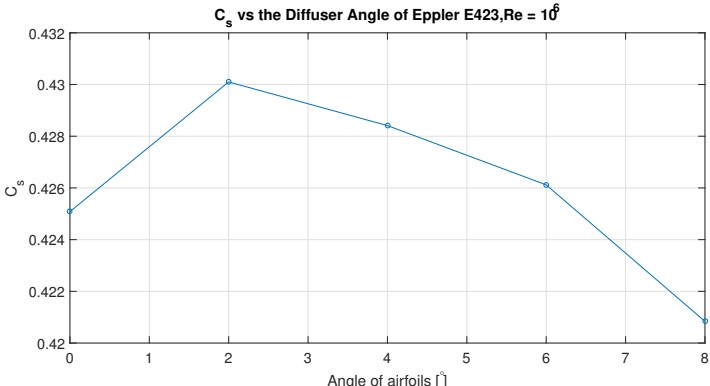

**Figure 3.** Eppler E423 vs Angles

## 3  Validation of Model

### 3.1  Validation of Rotor Model

The rotor was modeled as an actuator disc, which causes a pressure drop as the flow passes through by applying a thrust force to the flow. To do so, a second flow region which models the rotor part was created. The Star-CCM+ allows users to define how two flow regions interact with each other. In the model, this intersection was adjusted to apply the desired pressure drop value. The thrust and power coefficients were calculated by following Eq. 2 and Eq.3, Martin (2010).

$$C_T = \frac{\Delta_p A_{disc}}{0.5 A_{disc} V_0} \tag{2}$$

, where in the computations the $\Delta_p$ values were pre-defined and varied to derive the thrust coefficients.

$$C_{P,b} = C_T V_{disc} \tag{3}$$

The model was validated by comparing the simulation results with $C_p - C_T$ curve of a bare rotor as seen below at Fig.4 .

### 3.2  Validation of Diffuser Model

Before starting the design process, the developed CFD model and the calculation method were also validated by investigating
a previously done study, Hansen et al. (2000). As it is shown at Tab. 2, the discrepancy between two results is insignificant. The error is concluded to be related to the mesh size difference between two models use.



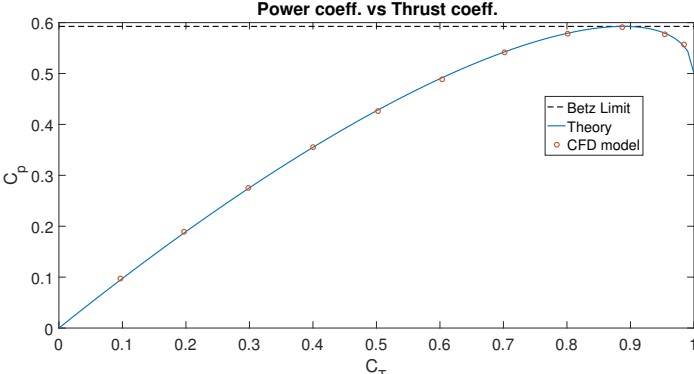

**Figure 4.** $C_p$ Comparison between Model and Theory

|  | Hansen O.L. | Present Work | Error |
|---|---|---|---|
| Max$C_{p,d}$ | 0.9 | 0.94 | 5% |
| $C_T$ | 0.89 | 0.87 | 2.17% |

**Table 2.** Optimum $C_p$-$C_T$ Values

## 4   Design and Evaluation

### 4.1   Design

CFD model had steady-flow conditions with $k - \omega$ SST turbulence model. $Re$ and the diffuser length-rotor diameter ratio were
set as $10^6$, which is corresponding to 7.1 [m/s] design wind speed and 0.2, respectively. An axisymmetric swirl could have been
used to mimic the rotation effect of the rotor inside the diffuser, which may have increased the performance by limiting the
separation of flow along the diffuser surface as it is concluded in Phillips et al. (2003). However, as the model results matched
with the bare rotor theory and Hansen Diffuser presented in Hansen et al. (2000), the swirl model was not applied. Design
parameters to be used in the diffuser design process were determined as, rotor gap ratio, which indicates the ratio of vertical
distance between rotor tip and the inner surface of diffuser over the rotor diameter, the angle of attack of airfoil, which is the
expansion angle of diffuser.

Additionally, for the second and third design options, the gap between two diffusers was included in the design parameters as
height and width ratios, which are vertical and horizontal lengths normalized with the length of second diffuser, respectively.
The horizontal distance between the diffuser end and the rotor was not included to the process as the maximum flow rate would
be at the point where the gap is minimum. The results from the initial studies also were aligned with that, as can be seen in
Fig.5.

The geometrical ratios of gurney flap which to be used in the third design was optimized separately based on the diffuser
length.



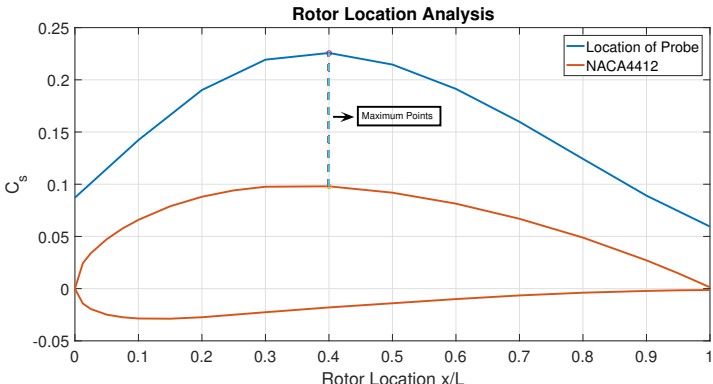

**Figure 5.** Diffuser Coefficient vs Probe Location

### 4.1.1 Augmentation Factor

In the design process, as the simulations were for loaded scenarios, diffuser coefficient, $C_s$ can not be used. Instead, augmentation factor which was used as a calculation method by Hansen et al. (2000), was chosen for the evaluation of each case.

$$C_{p,Diffuser} = \frac{P}{0.5 \, \rho \, V_0{}^3 A} = \frac{T V_{diffuser}}{0.5 \, \rho \, V_0^2 \, \frac{V_0}{V_{diffuser}} \, V_{diffuser} \, A} = C_{T,bare} \, \epsilon \qquad (4)$$

### 4.1.2 Thrust Coefficient, $C_T$

The flow separation along the diffuser surface is predominantly dependent on diffuser angle, rotor gap ratio and the thrust coefficient, which can be represented as $f(\alpha, C_T, D_r)$. If the rotor gap is higher than it should be, the higher thrust values may cause separation of flow at the trailing edge of diffuser, as it is shown in Fig. 6.

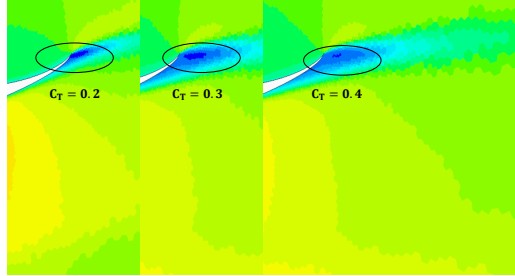

**Figure 6.** Flow separation depending on thrust coefficient due to the geometry of gap, S1223-$\alpha = 10$

However, if the gap is appropriately adjusted, higher thrust values would increase the flow rate passing through the gap as creating "jet effect", which subsequently would reduce the separation of flow from the diffuser surface.



### 4.1.3 Diffuser Design

Three design concepts were optimized by applying a basic design optimization algorithm. The algorithm displayed in Fig.7 is only to show the design methodology which was also used to obtain the second design option by replacing the rotor gap ratio with the second gap ratio, which is sub-grouped as the height and the width ratios. The second element diffuser was selected a

```
 1:  Initialize first diffuser angle, α.
 2:  Initialize rotor gap ratio, D_r.
 3:  Initialize counter, i = 1 .
 4:  for do j = 1 to 2
 5:      while C_p(i + 1) < C_p(i) do α(i + 1) = α(i) + 2
 6:          if Convergence criteria not met then
 7:              Break
 8:          end if
 9:      end while
10:      while C_p(i + 1) < C_p(i) do D_r(i + 1) = D_r(i) − 0.01 * L_rotor
11:          if Convergence criteria not met then
12:              Break
13:          end if
14:      end while
15:  end for                                          ▷ First Diffuser
```

**Figure 7.** Design Algorithm

thin airfoil from the list of airfoils determined, GOE-204, in parallel to the previous studies on multi-element diffusers,Phillips et al. (2003), Before adding the gurney flap on the second element to obtain the third design option, the rotor-diffuser length ratio was examined based on $C_p$ values, which are proportional to $L/D$. While the length ratio decreases, the diffuser effect weakens. However, it can not be concluded that increasing the length ratio does always result in an increase in power coefficient as the analysis showed that after a certain point, increasing the ratio results in a drop of the flow rate due to blockage effect.

### 4.1.4 Rotor Design

The modified Glauert Optimization Method which was adopted by Vaz and Wood (2016) for diffuser augmented wind turbines was used to obtain rotor designs.

The design wind speed, angle of attack, rotational speed, number of blades are 7.5 [m/s], 6.75 [deg], 480 [rpm], and 3 respectively. The rotors are based on a NACA0012 airfoil. Including a bare rotor to be used as a reference, four rotors were designed by using the calculated speed-up ratios of diffusers shown above at Fig.8

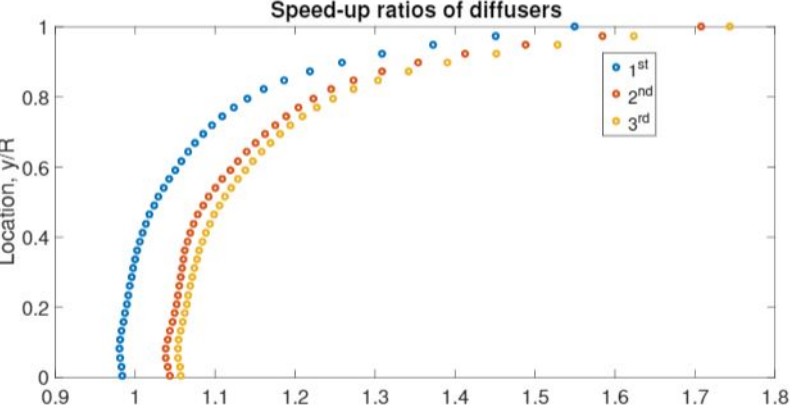

**Figure 8.** Speed-up ratios,$\epsilon = \frac{V_{rotor}}{V_\infty}$

Chord and twist distributions of optimized rotors are shown at Fig. 9 and Fig.10 respectively.

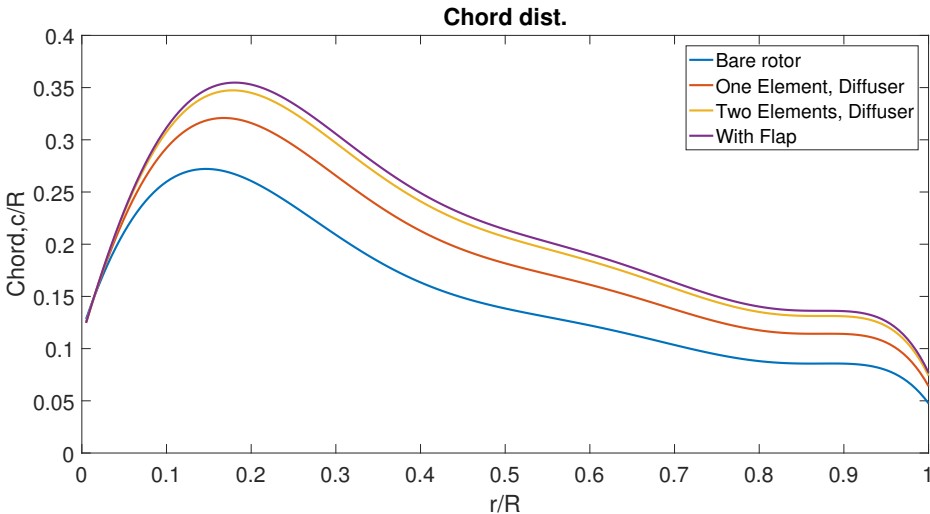

**Figure 9.** Chord distribution of blades





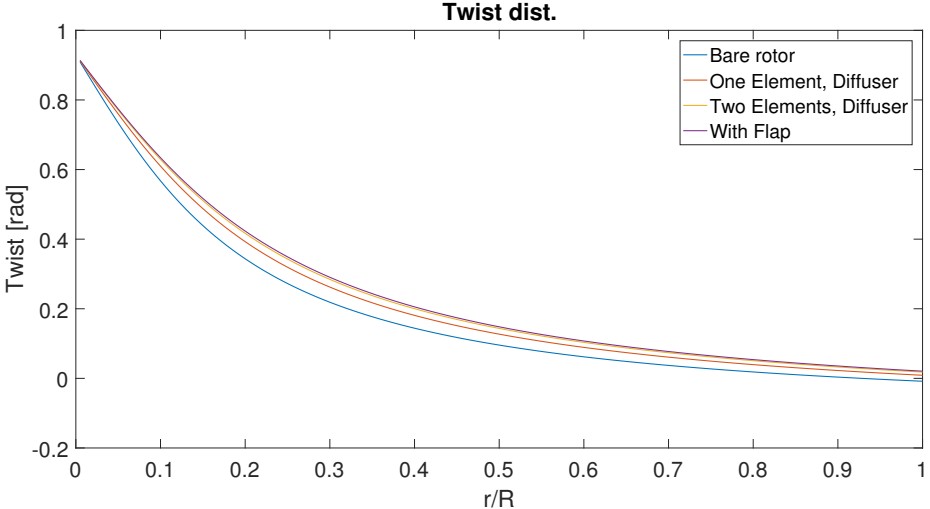

**Figure 10.** Twist distribution of blades

## 4.2 Evaluation

The evaluation of diffusers was done based on 2 cases; Case-1, which indicates the performance of diffusers with their correspondingly optimized rotor designs and Case-2, which evaluates the effect of diffusers on flow as if they were just put around an open rotor. The power outputs and the power coefficients were obtained by using an edited BEM code for diffuser augmented wind turbines presented at Tavares Dias Do Rio Vaz et al. (2014). However, this study made a small change on how the speed ratio is applied into the code by introducing it as a function of location, $\gamma(r)$, instead of keeping it as its maximum value on rotor section as proposed at Tavares Dias Do Rio Vaz et al. (2014). Both cases point out the single-element diffuser, as the optimum design as it is seen below in Fig. 11 and 12 . Even though the two-element diffuser also creates a significant increase in the power output in comparison to the bare turbine, the increased complexity of the design seems not to be worthy of upgrading the single diffuser with a second element in terms of relative power increase.

In case-2, as wind speeds that rotor faces ramp up due to the augmentation effect of the diffusers, a significant improvement is evident below the design wind speed. The diffuser effect also leads to the separation of flow and decrease in power extraction on higher wind speeds. However the results show that a modification of a bare rotor by adding a single-element diffuser could be a good option to increase the operating range of the turbine.

The evaluation of designs was also done by comparing the relative load increase with the power increase values , which were found to be higher than the total load increase; the rotor thrust, and the axial loading increase on diffuser surfaces. The third design option is concluded to be not such an applicable option when the increase on diffuser loading is considered.

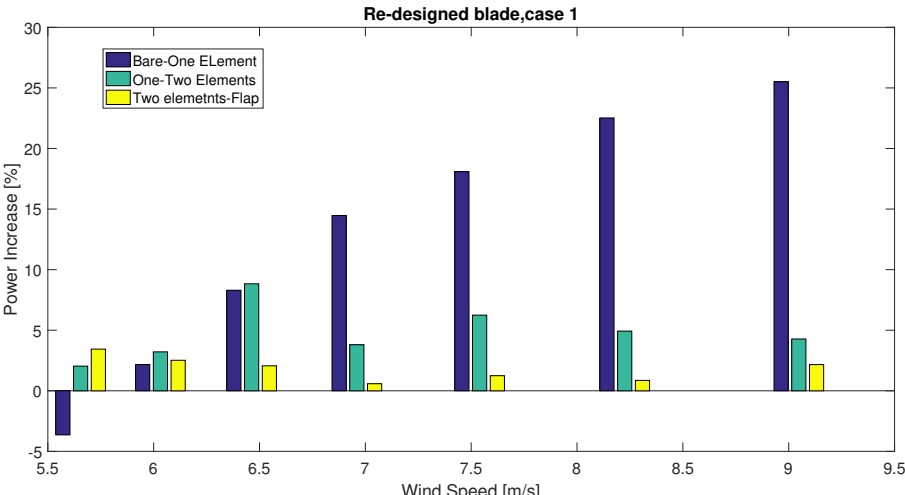

**Figure 11.** Relative Power Increase, Case 1

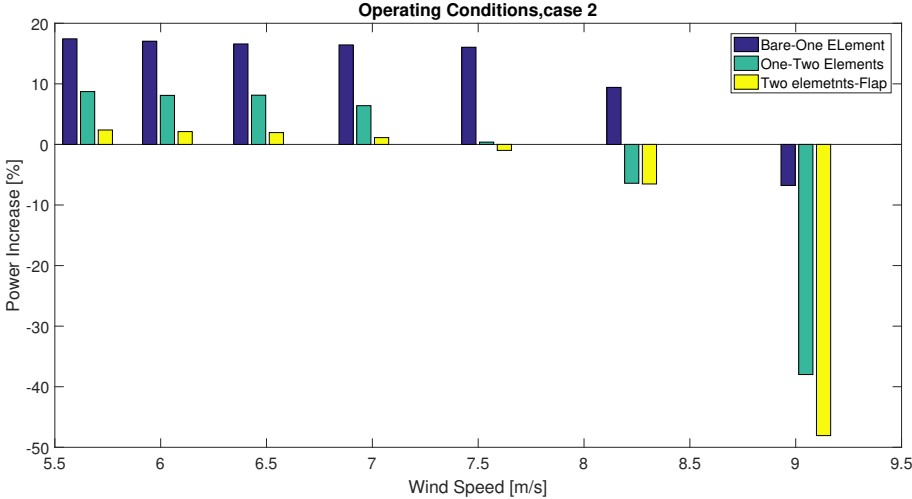

**Figure 12.** Relative Power Increase, Case 2

## 5  Conclusions

All three optimized diffuser designs were found to be capable of exceeding the Betz limit, $0.593$. However, the power coefficients calculated both from CFD and BEM have different values. The main reason is the difference in the calculation methodology.

For CFD results, the classical diffuser theory was used where it is assumed that thrust remains unchanged for a rotor with a



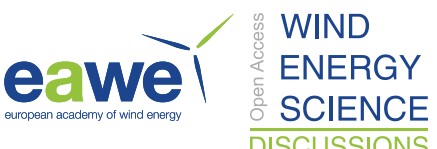

diffuser, which is not valid. The edited version of BEM does not require this assumption and indicates a relation between the speed-up ratio,($\gamma$) and the axial induction factor.

Additionally, it should also be noted that the speed-up ratio $\gamma$ values used in the calculations with the edited BEM have been already obtained from a loaded case simulation on CFD. Yet, the proposed edited BEM,Tavares Dias Do Rio Vaz et al. (2014), is found to be consistent within itself.

If the same process is repeated with a different design approach of rotor blades in the future, it should be taken into account that at zero loading conditions, the flow separates along the trailing edges of the optimized diffuser designs. Thus, different power values should be expected seeing when the BEM and 3-D CFD results are compared as the edited BEM does not address the flow separation inside the diffuser.

The diffuser coefficient proposed at Hjort and Larsen (2014) is concluded as an efficient method for the evaluation of different
airfoils in the design process. The same method would be useful to identify which airfoil profiles would be better on being used as a diffuser, based on geometrical parameters.

*Code availability.* None

*Data availability.* None



## Appendix A: Diffuser Geometries

Fig. A1 shows the final geometries of three diffusers. The symmetry axis goes from the origin which is not displayed in order
to give the dimensions of diffusers more prominence.

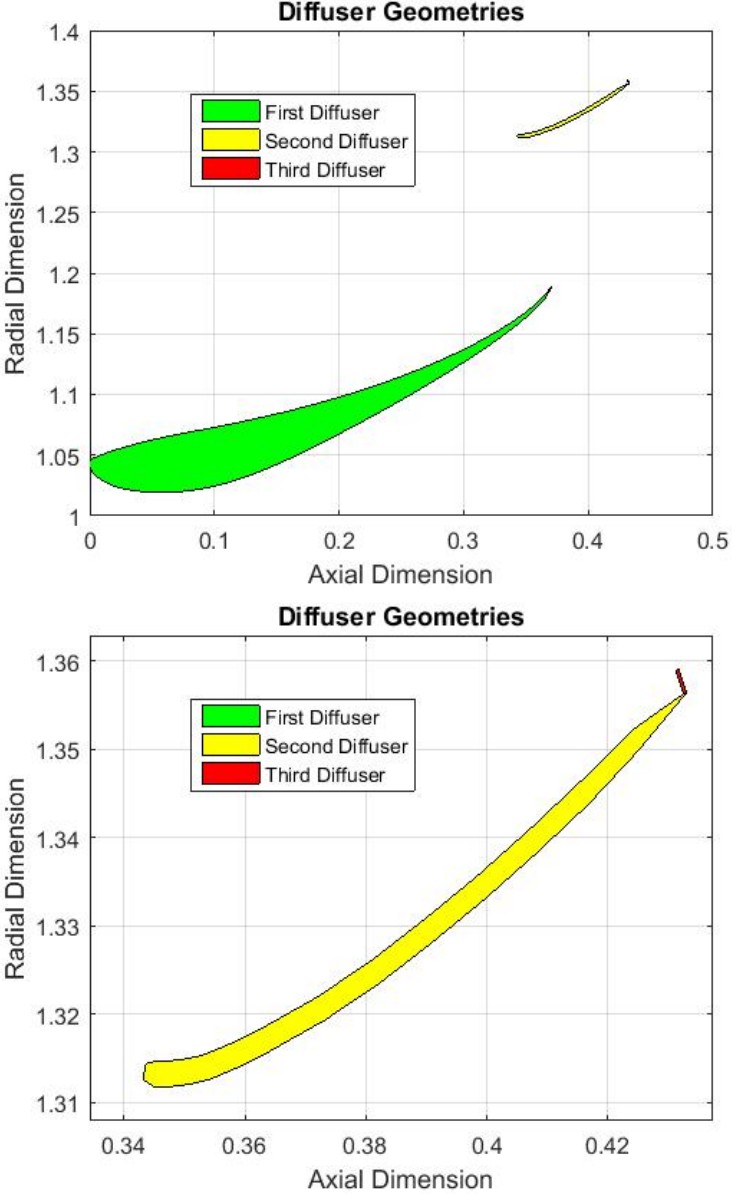

**Figure A1.** Diffuser Geometries



## A1 Gorney Flap Design Parameters

Fig. A2 and Tab.A1 indicate the optimized geometry parameters of Gurney Flap to be used on the third diffuser design. Width and Height values were normalized by the chord length.

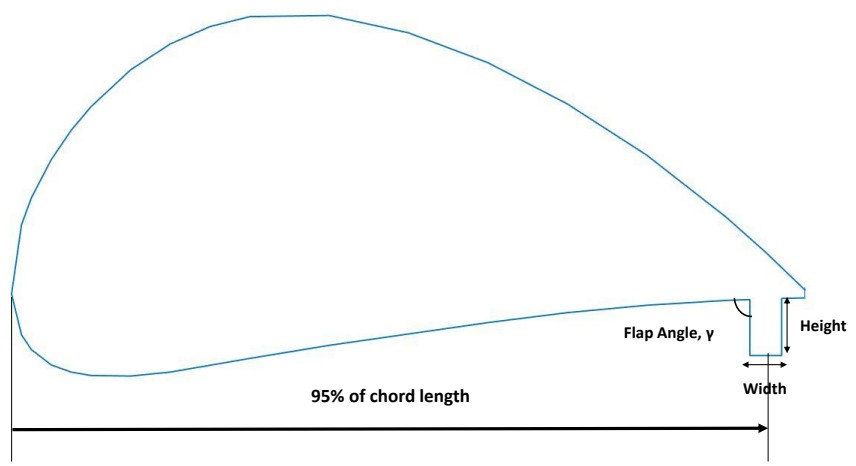

**Figure A2.**

| Width [%$c$] | Height [%$c$] | Flap Angle [deg] |
|---|---|---|
| 0.5 | 3 | 90 |

**Table A1.** Optimized Gurney-Flap Geometry



## A2 Diffuser Length - Rotor Diameter Ratio effect on $C_p$

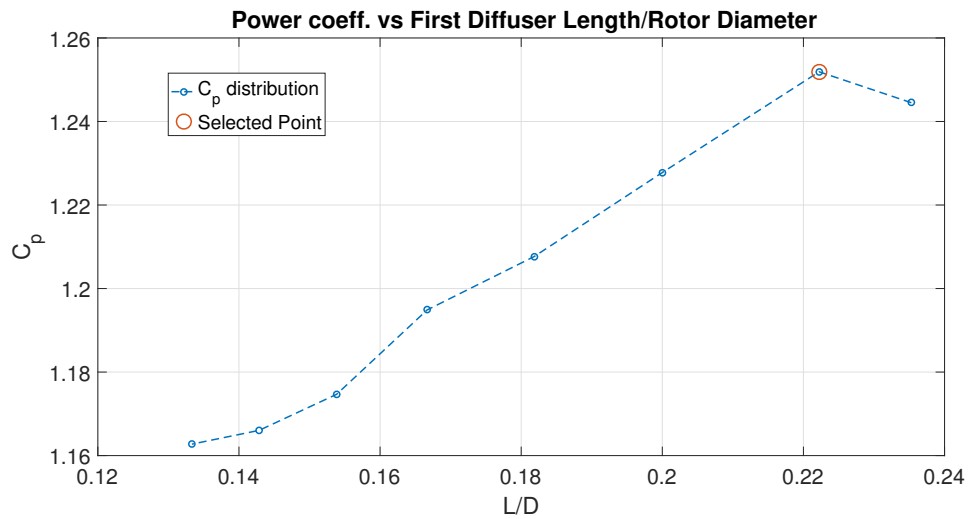

**Figure A3.** $L_1/D_{rotor}$ of the first diffuser

## Appendix B: Blade Element Theory with Diffuser

This section shows the equations of edited version of BEM for turbines with diffusers, which was presented at Tavares Dias Do Rio Vaz et al. (2014).

$$\gamma = \frac{V_{1,max}}{V_0} \tag{B1}$$

For different loading conditions, the velocity speed-up ratio was defined as a function of induction factor, shown in the Eq.B2.

$$\epsilon = \gamma\left(1 - a^*\right) \tag{B2}$$

$$\tan\phi = \gamma\frac{(1-a^*)V_0}{(1+a_1^*)\Omega r} \tag{B3}$$

$$C_T^d = \frac{T_d}{\frac{1}{2}\rho A V_0^2} = \frac{\frac{1}{2}\rho A\left(V_0^2 - V_4^2\right)}{\frac{1}{2}\rho A V_0^2} = 4a^*\left(1 - a^*\right) \tag{B4}$$



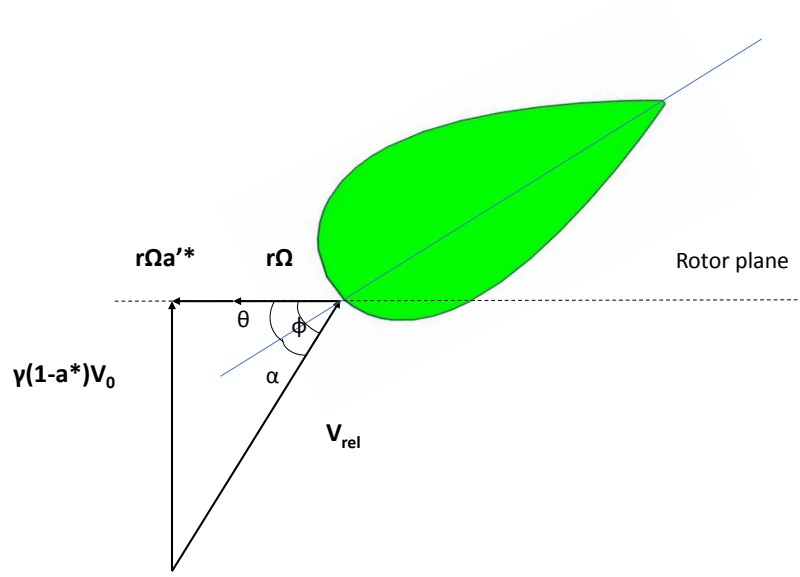

**Figure B1.** Velocity triangle with diffuser

$$C_T^d = \Big(\frac{V_1}{V_0 \, sin(\phi)}\Big)^2 \sigma \, C_n \tag{B5}$$

$\quad$ $$C_T^d = \frac{T_d}{\frac{1}{2}\,\rho\,A\,V_0^2} = \frac{\frac{1}{2}\,\rho\,A\,(V_0^2 - V_4^2)}{\frac{1}{2}\,\rho\,A\,V_0^2} = 4\,a^*\,(1 - a^*) \tag{B6}$$

$$C_T^d = \Big(\frac{V_1}{V_0 \, sin(\phi)}\Big)^2 \sigma \, C_n \tag{B7}$$

$$\frac{a^*}{1 - a^*} = \gamma^2 \, \frac{\sigma \, C_n}{4 \sin^2(\phi)} \tag{B8}$$

$$\frac{a'^*}{1 + a'^*} = \frac{\sigma \, C_t}{4 \, sin(\phi) \, cos(\phi)} \tag{B9}$$





$$C_n = C_l \cos\phi + C_d \sin\phi \quad C_t = C_l \sin\phi - C_d \cos\phi \tag{B10}$$

$$C_p^d = \gamma \, 4 \, a^* \left(1 - (a^*)^2\right) \tag{B11}$$

$$C_T^d = \begin{cases} 4 \, a^* \left(1 - a^*\right) F & a^* \leqslant \frac{1}{3} \\ 4 \, a^* \left(1 - \frac{a^*}{4} \left(5 - 3 \, a^*\right)\right) F & a^* > \frac{1}{3} \end{cases} \tag{B12}$$

$$a^* = \begin{cases} \frac{1}{1+K} & a^* \leqslant a_c \\ 0.5 \left(2 + K \left(1 - 2 \, a_c\right) - \sqrt{\left(K \left(1 - 2 \, a_c\right) + 2\right)^2 + a \left(K \, a_c^2 - 1\right)}\right) & a^* > a_c \end{cases} \tag{B13}$$

$$K = \frac{4 \, F \, sin^2(\phi)}{\gamma^2 \, \sigma \, C_n} \tag{B14}$$

**Appendix C: Drag Coefficients of diffuser designs**

| Design Case | $C_d$ |
|---|---|
| One element | 0.6140 |
| Two elements | 0.6202 |
| Flap | 0.6676 |

**Table C1.** Drag coefficients of diffuser surface

*Author contributions.* E.V. gathered data, reviewed literature, performed analysis and design steps and wrote the paper

*Competing interests.* The author declares that he has no conflict of interest



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
