# Peer review of "CFD based design of diffuser augmented wind turbines"

_Wind Energy Science, 2019_

## Referee Comment (RC1) · Anonymous Referee #1 · 17 Dec 2019

WES2019ãĂĂMS No.: wes-2019-75 Special Issue: Wind Energy Science Conference 2019

Title : CFD based design of diffuser augmented wind turbines

Author(s) : Ertem Vehid

Referee's Comments :

This paper describes a numerical study of the flow around a diffuser augmented wind turbine. The author used a CFD tool provided by Star- CCM+. Although the present paper is a study using CFD, there is no explanation in detail for the method. However, comparing the results between BEM and the present CFD, the author says that the difference in the results comes from methodology. Before reviewing the results and discussions, there are a lot of problems as a scientific paper, because there are no

explanation about the symbols and the symbols are not unified for the whole paper. It is very hard to read and understand the paper, this is not comprehensive one. For the overall evaluation, the present paper seems to give relatively good numerical results. However, a couple of discussions are insufficient. The referee described those problems below in the detailed points. It seems to the referee that a further improvement and revise still remain for the present paper. If the author can revise largely, the present paper should be accepted for the WES under the present results.

Major points:

For the numerical method, what kind of grid type did the author employ? What is the grid resolutions around the turbine and diffuser? How is large for the computational domain and how is the blockage? Did the author pay attentions to the Reynolds number effect and the grid resolution dependence to make clear the flow characteristics around the ducted wind turbine which shows flow separation and reattachment inside the diffuser, and vortex shedding from the diffuser. It seems that the flow around the ducted turbine both inside and outside of the duct are highly unsteady, unstable and turbulent flows. The reviewer cannot understand the accuracy of CFD presented in this paper.

Furthermore, it is necessary to correct the following points.

Detailed points

1. p. 2, Figure 1; Please describe the definitions of all the symbols in the present paper. A? U? p? What are Ad(i)? and Ud(i) in equation (1)

2. p. 3, line 1; What is the diffuser angle? And What is Re? For Re, what is the reference length?

3. P.4, line 64; What is V0 (V0*V0?) V0 is equl to $U_\infty$ or Ud? What is $\Delta p$?

4. And more

Please also note the supplement to this comment:
https://www.wind-energ-sci-discuss.net/wes-2019-75/wes-2019-75-RC1-
supplement.pdf

---

## Referee Comment (RC2) · Anonymous Referee #2 · 13 Apr 2020

Review of
Title: CFD based design of Diffuser Augmented Wind Turbines (DAWT)

By: Ertem Vehid

Short résumé
CFD on diffusor augmented wind turbines…

Specific comments:
1. The DAWTs are suggested as promising in regards to the cost of energy. The arguments for this are not strong and I think the author could be more skeptical in presenting the economic future for DAWT. Overall I disagree with the statement and I have not seen data that suggest otherwise. The cost of the DAWT should in my opinion also be compared to the cost of an extended wind turbine blade solution having the same power as a DAWT. Extending the blades compared to the expense of the diffusor would not be to the advantage of DAWT considering LCOE. Comparing DAWT to large modern wind turbines is also problematic as upscaling of DAWT to comparable size or MW power is in my view not realistic and misleading not to underline. On the other hand, the DAWT could have a future in its own right and could supplement the marked for small to moderate size wind turbines.
2. The introduction and abstract both starts motivating with reference to loose economic arguments, e.g, lowering energy cost, a trend seen now days in too many papers related to renewable energy. I am not a big fan of this development. I prefer a strong technical motivation for why this paper is relevant. DAWT has been around since the 1960-70 and in that aspect it is also important to underline what is new in the present paper.
3. The intro could include some status on what has been achieved by others doing CFD on DAWT. Many studies with CFD on DAWT have been carried out the last 2 decades, what is the status and what is new in the present paper?
4. Foil selection: this sections needs more work, what are the main achievements reported by the 3 referenced works?
5. The physical definition of the diffusor coefficient is unclear in the text. Eq.1 states it is the mean speed up increase at the most narrow point… coefficient for mass flow increase? More like a measure than a model…
6. CFD setup: some key details should be included e.g. axisymmetric or 3D grid, turbulence model used, number of cells, etc…
7. The data in table 1 has mixed info: The foils mentioned should include the lift at zero AOA to make the table more meaningful for selecting/evaluating foil to choose.
8. The sections headlines in are not well thought through
9. How is chord/radius ratio chosen initially?
10. It is not clear that the diffusor coefficient should be maximized for the no load situation. Why?
11. Figure 3. Why is the curve for S1223 not on this graph?
12. Section 3.1: what does $V_{disc}$ look like?
13. Eq.2, divide with Vo or Vo^2?
14. Sec.3.2, this section is unclear. Fig.4 shows comparison with axial momentum theory for integral CT, CP, local velocities could be shown also. Table 2, Max CP=0.9? Bare rotor or with diffusor?
15. More details on the CFD used introduced in sec.4. This is a bit backward since the CFD was used previously with no details. The aspect ratio of 0.2 is first mentioned here, it could be more chronological.
16. Swirl is omitted, Ok simplification, on the other hand, the BEM Eqs. in app. B includes a'. Including tangential loading should be manageable and provide a shaft CP.
17. A figure explaining the geometry definition of designs clearly is preferred, sec.4.1 is unclear.
18. Augmentation factor could be explained e.g. as a power factor for DAWT. Computing Cs is also interesting I think.

19. The area A in Eq.4 is not defined. Some dispute is associated with the choice of A as choosing the rotor area, Ar, gives a very high CP. On the other hand, the largest area may also be used as reference e.g. the exit diffusor area Ae. In my opinion this area should be used as this is the area the turbine effectively covers. As minimum, the area ratio Ae/Ar should be stated. Likewise the size of a bare rotor that can produce the same power, as stated earlier.
20. The gap investigated, what is the recommended gap? Is the Jet effect wanted or should the gap clearance be a small as possible?
21. The optimization method is characterized more like a parameter variation. No gradients are computed…
22. Sec.4.1.3: It is not easy to follow changes and choices of designs in this section, it is multiple changes and some ordering would help, e.g. a table or figure.
23. Figure 8. Figure text is missing, as for most figures throughout the paper… what is $1^{st}$, $2^{nd}$ and $3^{rd}$? Reference to App. needed.
24. Fig. 9. The chord distribution for the bare rotor seems bumpy. Why is the modification to the optimized Glauert design resulting in this for the bare rotor?
25. What does the loading look like?
26. Sec. 4.2. Also a confusing section. The gamma function is introduced in this section, but this should have been explained earlier.
27. Figures 11 and 12 need comments.
28. Conclusions: The diffusors design can exceed the Betz limit, yes, but what is new? Hjort et al. and others has shown this before,

App.
1. A1. Is figure A2 a NACA4412? Please plot geometry in scale 1:1 for x-y cords. Also fig. 5
2. B1. The Eq. in app. B needs comments.

Concluding remarks:

The present text is not easy to follow. The introduction should include a bit of review I think. Stating that the purpose of the paper is to "maximize diffusor power output" is somewhat too simplified, as the intro does not reference prior efforts to do this and what has been achieved. In what way has this not been addressed in prior work? What has been achieved? Is a simple parameter variation enough to get a better design?

The text is in general not clear in my opinion, the paper needs restructuring and clarity. It would be difficult for anyone to redo what has been computed by the author, it is difficult to follow chronological what has been done. All figures are without sufficient explanatory text about what exactly is shown.

I cannot recommend the present text for publication.

Wording:
Betz-Jawceskesy - Betz-Joukowsky

Gorney - Gurney

Many wording issues, the written English needs a careful read through.

---

## Editor Comment (EC1) · Katherine Dykes (Editor) · 15 Apr 2020

Thank you for your submission. I believe there are some fundamental issues with this paper describing CFD-based design with the diffuser coefficient as an evaluation metric. However, the relationship between diffuser coefficient and the performance of a rotor+diffuser combination has not been made clear. Improvements to rotor performance following the application of the modified Glauert Optimization Method has not been discussed. No power coefficients have been shown for the diffuser optimized designs, but in the conclusions the author claims that all designs can exceed Betz; however, he also points out that these power values should be different than 3-D CFD because they neglect the established issue of flow separation. In reality, accounting for separation should actually reduce performance.

Moreover, the paper lacking in many details and discussion, including those regarding

the CFD model despite the paper title emphasizing a CFD focus. The CFD airfoil results have not been sufficiently validated and the connection to the full rotor design is unclear—the initially selected optimal airfoil (S1223) is later replaced with a NACA0012 design. The introduction does not properly motivate the present work (the author states that a main design challenge is flow separation, which has been neglected in the present study) or choice of design concepts, and has no references to recent work. Please see the attached annotated PDF for additional comments.

Please also note the supplement to this comment: https://www.wind-energ-sci-discuss.net/wes-2019-75/wes-2019-75-EC1-supplement.pdf

**Supplement:**

[revised manuscript text omitted]

---

## Author Comment (AC1) · 10 May 2020

The author expresses his gratitude to all referees and additional commenters who were kind enough to review and share their valuable thoughts about the current work. The author also understands the concerns and the critics on how the study was presented in this paper. These will be crucial in the rewriting process of this work as well as for future works.
* * *